# ROYAL SOCIETY
# OPEN SCIENCE

health and disease and epidemiology/evolution/genetics

cervid, chronic wasting disease, natural selection, *Odocoileus hemionus*, prion protein gene, wildlife disease

**Author for correspondence:**
Holly B. Ernest
e-mail: holly.ernest@uwyo.edu

# Spatio-temporal analyses reveal infectious disease-driven selection in a free-ranging ungulate

Melanie E. F. LaCava[1], Jennifer L. Malmberg[3],
William H. Edwards[4], Laura N. L. Johnson[2],
Samantha E. Allen[5] and Holly B. Ernest[1]

[1]Wildlife Genomics and Disease Ecology Laboratory, Department of Veterinary Sciences, Program in Ecology and [2]Wildlife Genomics and Disease Ecology Laboratory, Department of Veterinary Sciences, University of Wyoming, Laramie, WY 82071, USA
[3]Department of Veterinary Sciences, Wyoming State Veterinary Laboratory, University of Wyoming, Laramie, WY 82070, USA
[4]Wyoming Game and Fish Department, Wildlife Health Laboratory, Laramie, WY 82070, USA
[5]Wyoming Game and Fish Department, Department of Veterinary Sciences, University of Wyoming, Laramie, WY 82070, USA

(iD) MEFL, 0000-0001-7921-9184; JLM, 0000-0001-9066-721X;
LNLJ, 0000-0002-1130-8524; HBE, 0000-0002-0205-8818

Infectious diseases play an important role in wildlife population dynamics by altering individual fitness, but detecting disease-driven natural selection in free-ranging populations is difficult due to complex disease–host relationships. Chronic wasting disease (CWD) is a fatal infectious prion disease in cervids for which mutations in a single gene have been mechanistically linked to disease outcomes, providing a rare opportunity to study disease-driven selection in wildlife. In Wyoming, USA, CWD has gradually spread across mule deer (*Odocoileus hemionus*) populations, producing natural variation in disease history to evaluate selection pressure. We used spatial variation and a novel temporal comparison to investigate the relationship between CWD and a mutation at codon 225 of the mule deer prion protein gene that slows disease progression. We found that individuals with the 'slow' 225F allele were less likely to test positive for CWD, and the 225F allele was more common in herds exposed to CWD longer. We also found that in the past 2 decades, the 225F allele frequency increased more in herds with higher CWD prevalence. This study expanded on previous research by analysing spatio-temporal patterns of individual and herd-based disease data to present multiple lines of evidence for disease-driven selection in free-ranging wildlife.

# 1. Introduction

Infectious diseases pose a significant threat to global biodiversity and require extensive research and resources to manage [1–3]. Documenting selection pressure imposed by infectious diseases in natural systems remains challenging due to complex disease–host relationships and the myriad factors influencing host fitness. Previous research has demonstrated a relationship between wildlife disease phenotypes and diversity in immune-related genes such as the major histocompatibility complex (MHC) [4–6], with less focus on variation in other genes [7]. Methods such as genome-wide association studies are commonly applied to identify putative loci under selection [8], but the identification of mechanistic links between disease phenotypes and specific mutations in the genome remains limited in free-ranging animal populations [7]. Chronic wasting disease (CWD) is an infectious prion disease in captive and free-ranging cervids for which mutations in a single gene have been linked to variation in disease outcomes in affected species [9], providing a rare opportunity to study disease-driven selection in free-ranging wildlife.

CWD is caused by the conversion of normal cellular prion proteins ($PrP^C$) into misfolded, protease-resistant prion proteins ($PrP^{CWD}$) that accumulate in the central nervous system and cause spongiform encephalopathy, ultimately resulting in death [10,11]. Several studies have identified mutations in the prion protein (PRNP) gene that alter the tertiary structure of the PrP protein, which reduces the conversion rate of $PrP^C$ to $PrP^{CWD}$ and results in slower disease progression [12–14]. For example, experimentally inoculated mule deer (*Odocoileus hemionus*) with a rare mutation at PRNP codon 225 developed clinical CWD after approximately 3 years, whereas mule deer without this mutation developed clinical CWD is less than 2 years [15]. So far, research has documented variation in disease progression related to PRNP genotypes in mule deer, white-tailed deer (*Odocoileus virgianus*), elk (*Cervus elaphus nelsoni*) and Eurasian reindeer (*Rangifer tarandus*), with no evidence of a completely resistant genotype [9,16]. Slower disease progression could allow more opportunities for individuals to reproduce before succumbing to CWD, and thus drive selection in favour of 'slow' PRNP genotypes.

Demographic models that incorporated genotype-specific fitness led to higher population growth estimates for individuals with slow PRNP genotypes [17–19]; therefore, selection favouring slow PRNP genotypes could markedly affect long-term population dynamics. Since its discovery in northern Colorado, USA, in the 1960s [20], CWD has expanded its range in North America and has spread to northern Europe and southeast Asia [21,22], and CWD has contributed to population declines in mule deer and white-tailed deer [19,23–25]. The geographical expansion of CWD combined with its impact on population growth led to the inclusion of CWD in a list of the top 15 emerging issues for global conservation and biological diversity in 2018 [26]. Characterizing evidence of natural selection related to CWD could improve projections of population dynamics as CWD continues to threaten cervid populations globally.

CWD has gradually spread across Wyoming, USA, over the past several decades, leading to spatial variation in disease history which can be used as a proxy for change over time to evaluate evidence of natural selection [27]. In addition, previously published CWD prevalence and PRNP genotype frequencies for mule deer herds in southeastern Wyoming [15] provide an opportunity to evaluate actual change in genotype frequencies across multiple herds over time. In mule deer, two PRNP codons, 20 and 225, have been associated with disease phenotypes. At PRNP codon 225 in mule deer, the reference sequence (TCC) encodes the amino acid serine (S) and the variant sequence (TTC) encodes the amino acid phenylalanine (F) [28]. Mule deer possessing at least one F allele at codon 225 experienced delayed accumulation of $PrP^{CWD}$ in the central nervous system [29] and delayed development of clinical CWD [15,30]. Experimental inoculation of transgenic mice with the mule deer PRNP gene demonstrated the same clinical variation related to codon 225 genotypes, providing further support that the mutation at codon 225 causes the observed phenotypic variation [12]. Jewell *et al*. [15] found that individuals with the slow 225F allele were less likely to test positive for CWD. A later study documented a 10% increase in the slow 225F allele frequency in less than 10 years [19], although this study focused on a single mule deer herd and, therefore, the change in genotype frequency could potentially be due to genetic drift, rather than selection. Other studies have found a correlation between CWD status and PRNP codon 20 genotype in mule deer in western Canada and Nebraska, USA [31,32]; however, clinical evidence for differential disease progression related to codon 20 does not yet exist. These studies found that at codon 20, mule deer with two copies of the common amino acid aspartic acid (D; nucleotide sequence GAC) were less likely to test positive for CWD than deer with at least one copy of the variant amino acid glycine (G; nucleotide sequence GGC).

We used spatial variation in CWD history and a novel temporal comparison across multiple herds to evaluate the hypothesis that CWD is driving selection in free-ranging mule deer in Wyoming, USA. We evaluated three predictions related to PRNP codons 20 and 225:

1. Individuals possessing the slow allele will be less likely to test positive for CWD.
2. The slow allele will be more common in herds with longer exposure to CWD.
3. The slow allele will increase more over time in herds with higher CWD prevalence.

# 2. Methods

## 2.1. Disease data

The Wyoming Game and Fish Department (WGFD) Wildlife Health Laboratory provided individual CWD status (CWD+ or CWD−), herd CWD prevalence and year of first detection of CWD for this study. Individual CWD status was determined using an enzyme-linked immunosorbent assay (ELISA) [18,33,34]. The WGFD estimated CWD prevalence (i.e. the proportion of individuals that tested positive for CWD) for adult deer (males and females, though most harvested deer were males) based on ELISA tests in each herd from 2015 to 2019 (table 1). We used a single CWD prevalence estimate based on both sexes combined for each herd so that our current prevalence estimates would be comparable with historical estimates that did not specify the sex of sampled deer [15]. Mule deer herds have been defined by the WGFD as groups of deer using distinct geographical areas and having limited interchange with other herds [35]. CWD was first documented in free-ranging mule deer in Wyoming in 1985 [36], and the WGFD provided us with CWD surveillance programme data indicating the year they first detected CWD in each mule deer herd, beginning in 1992 (table 1). CWD surveillance began in different years in different herds, so the year of first detection is not a perfect corollary for the year CWD first arrived in each herd; however, surveillance expanded throughout the state as CWD spread, so the year of first detection represents an approximation for the relative length of time that herds have been exposed to the disease. We used samples from mule deer throughout the state of Wyoming, but prioritized samples from herds in three geographical regions with different CWD histories. These 14 focal herds included seven herds in southeastern Wyoming, where WGFD first detected CWD in mule deer in 1992 and CWD prevalence is highest in the state; four herds in northcentral Wyoming where CWD has been documented since 2003 and CWD prevalence is intermediate; and three herds in western Wyoming where CWD was first detected in 2014 and CWD prevalence is low or non-existent (figure 1 and table 1). In addition to variation in disease history, these three regions represent three genetic clusters of mule deer in the state [37].

## 2.2. Genetic sample collection

We used a combination of lymph node and blood samples to obtain genetic information for this study. The WGFD collected retropharyngeal lymph nodes from hunter-harvested mule deer in 2014–2016 for a CWD surveillance programme and later donated these samples to our laboratory. The WGFD recorded collection date, hunt area and sex of the deer upon sample collection. The WGFD also donated whole blood samples collected from mule deer captured in 2017–2019. Within one week of collection, we fractionated whole blood by centrifuging samples at $1000g$ for 10 min so we could isolate the buffy coat layer for use in DNA extraction.

We selected samples for DNA extraction based on sex (aiming for equal representation of males and females) and geographical location (aiming to include deer throughout the state of Wyoming while also prioritizing samples from the 14 focal herds). We extracted DNA from a total of 1261 samples (735 males and 526 females; 1069 lymph nodes and 192 blood samples).

## 2.3. DNA extraction, sequencing and genotyping

We extracted DNA using Qiagen DNeasy Blood and Tissue kits (Qiagen, Valencia, CA, USA) according to the manufacturer's instructions. We amplified 745 bp of the PRNP gene, including the open reading frame, using the primers described by Jewell *et al.* [15] with modifications to their polymerase chain reaction (PCR) conditions. For each 25 µl PCR reaction, we used 11.5 µl of 1× AmpliTaq Gold 360 MasterMix, 9 µl nuclease-free water, 1.5 µl of 10 µM forward primer (MD582F, 5′-ACATGGGCATATGATGCTGACACC-3′), 1.5 µl of 10 µM reverse primer (MD1479RC, 5′-ACTACAGGGCTGCAGGTAGATACT-3′) and 1.5 µl

**Table 1.** Summary of CWD and PRNP genotype data for 14 focal herds, including the year CWD was first detected by WGFD, the number of CWD ELISA tests in 2015–2019, CWD prevalence in 2015–2019, the number of samples with PRNP genotypes in 2014–2019, PRNP codon 225*F genotype frequencies in 2014–2019, CWD prevalence calculated in 2001–2003 (from Jewell et al. [15]) and PRNP codon 225*F genotype frequencies in 2001–2003 (from Jewell et al. [15]).

| herd name | first year CWD detected | CWD ELISA tests 2015–2019 | CWD prevalence 2015–2019 | samples genotyped 2015–2019 | 225*F frequency 2014–2019 | CWD prevalence 2001–2003 | 225*F frequency 2001–2003 |
|---|---|---|---|---|---|---|---|
| Laramie Mountains | 1992 | 556 | 0.223 | 72 | 0.208 | 0.200 | 0.067 |
| Goshen Rim | 1998 | 220 | 0.363 | 67 | 0.133 | 0.064 | 0.069 |
| Bates Hole/Hat Six | 1998 | 272 | 0.250 | 75 | 0.164 | 0.116 | 0.070 |
| South Converse | 1999 | 130 | 0.367 | 39 | 0.256 | 0.247 | 0.020 |
| Baggs | 2000 | 520 | 0.063 | 65 | 0.000 | 0.004 | 0.010 |
| Sheep Mountain | 2002 | 188 | 0.077 | 81 | 0.077 | 0.015 | 0.075 |
| Platte Valley | 2002 | 275 | 0.066 | 52 | 0.062 | 0.008 | 0.038 |
| Paintrock | 2003 | 293 | 0.129 | 69 | 0.043 | | |
| Southwest Bighorn | 2003 | 305 | 0.128 | 71 | 0.056 | | |
| Upper Powder River | 2004 | 279 | 0.134 | 87 | 0.069 | | |
| North Bighorn | 2008 | 299 | 0.061 | 74 | 0.014 | | |
| South Wind River | 2014 | 214 | 0.000 | 67 | 0.030 | | |
| Wyoming Range | 2016 | 362 | 0.000 | 70 | 0.057 | | |
| Sublette | 2017 | 307 | 0.006 | 76 | 0.053 | | |
| statewide total | | | | 1156 | 0.078 | | |

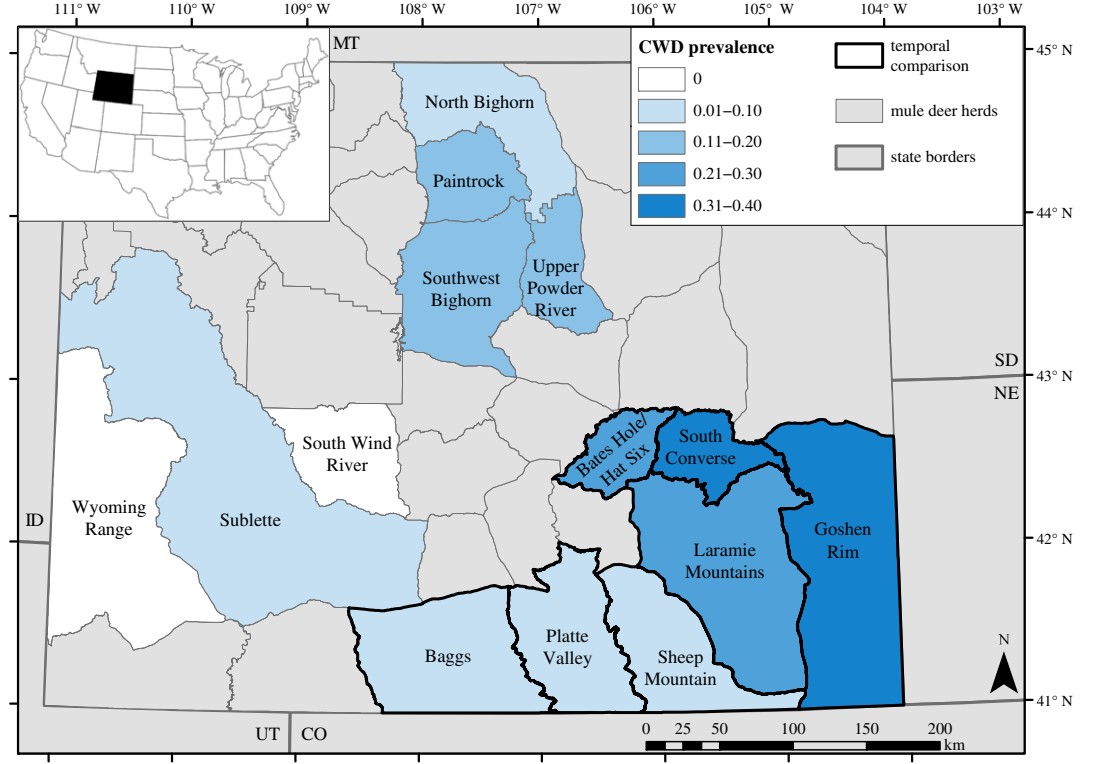

**Figure 1.** Map of our focal mule deer herds within the state of Wyoming, USA. Inset map shows the location of Wyoming within the USA and surrounding states are labelled by their two-letter code (CO, Colorado; ID, Idaho; MT, Montana; NE, Nebraska; SD, South Dakota; UT, Utah). Herds are coloured by CWD prevalence estimated in 2015–2019, and herd names are listed to correspond with data in table 1. The herds with thick black outlines were included in our temporal comparison. The majority of samples used in this study were collected in the 14 focal herds shown (*n* = 965); however, we genotyped 191 additional samples from other parts of the state, which we used in analyses not requiring herd CWD prevalence data.

extracted DNA. PCRs were conducted on Applied Bioscience SimpliAmp thermal cyclers with the following conditions: initial hold of 95°C for 7 min followed by 30 cycles of 95°C for 30 s, 62°C for 30 s and 72°C for 1 min, finished with a final hold of 72°C for 7 min. Amplification was verified by running PCR products on 1.5% agarose gels with 1× SYBR Safe in 0.5× TBE for 45 min at 115 V. PCR products were frozen immediately after PCR to limit the degradation of crude PCR product. We shipped PCR products on dry ice to Functional Biosciences, Inc. (Madison, WI, USA) for Exo SAP-IT purification and sequencing bidirectionally using the PCR primers. In addition, we sequenced 5% of samples a second time to confirm genotypes.

We aligned sequences to a mule deer PRNP reference sequence (GenBank Accession: AY330343.1) [28], edited sequences and verified base calls in Sequencher 5.4.1 (Gene Codes Corporation, Ann Arbor, MI, USA). We required two strands of sequencing to call a genotype. We analysed diploid genotypes rather than allele frequencies to avoid artificially inflating sample sizes and potentially biasing significance statistics [15,38]. Because the minor allele at codons 20 and 225 are both rare (minor allele frequency less than 0.1), we combined heterozygote genotypes with genotypes homozygous for the rare codon in each case, thereby analysing two genotype categories (20DD versus 20*G, 225SS versus 225*F) rather than three genotype categories [15,31].

## 2.4. Prediction 1: individuals possessing the slow allele will be less likely to test positive for CWD

With our first prediction, we evaluated whether an individual's disease status was related to their PRNP genotype at codon 20 or codon 225. We predicted that individuals possessing the slow allele at either site will be less likely to test positive for CWD because one aspect of slower disease progression may be a longer lag time between exposure to CWD and accumulation of detectable levels of infectious prions in deer with the slow allele [9]. We used logistic regression in the R package *stats* v. 4.0.2 [39]. Our null model related individual CWD status (i.e. CWD+ or CWD−) to CWD prevalence in the herd

**Table 2.** List of models tested for each of our three predictions. For our first prediction, we used logistic regression to relate individual CWD status (CWD+ or CWD−) to herd CWD prevalence (HerdPrev), sex, PRNP codon 225 genotype (codon225) and PRNP codon 20 genotype (codon20). For our second prediction, we used binomial regression with a complementary log–log link function to relate the frequency of the 225*F genotype (225*F) to the year CWD was first detected in each herd (YrsSinceDetect). For our third prediction, we used linear regression to relate the change in frequency of the 225*F genotype from 2001–2003 to 2015–2019 (Change225*F) to herd CWD prevalence calculated in 2001–2003 (HerdPrev_01_03) or in 2015–2019 (HerdPrev_15_19). We report AIC and ΔAIC if we performed model comparison.

| prediction | models | AIC | ΔAIC |
|---|---|---|---|
| #1 | CWD + ~HerdPrev + sex + codon225 + codon20 | 595.8 | 0 |
|  | CWD + ~HerdPrev + sex + codon225 | 597.5 | 1.7 |
|  | CWD + ~HerdPrev + codon225 | 606.4 | 10.6 |
|  | CWD + ~HerdPrev + sex + codon20 | 632.3 | 36.5 |
|  | CWD + ~HerdPrev + sex | 634.3 | 38.5 |
|  | CWD + ~HerdPrev + codon20 | 638.6 | 42.8 |
|  | CWD + ~HerdPrev | 640.7 | 44.9 |
|  | CWD + ~Region | 644.5 | 48.7 |
| #2 | 225*F ~ YrsSinceDetect | — | — |
|  | 20DD ~ YrsSinceDetect | — | — |
| #3 | Change225*F ~ HerdPrev_01_03 | −23.2 | 0 |
|  | Change225*F ~ HerdPrev_15_19 | −16.8 | 6.6 |

from which the animal was sampled. Herd CWD prevalence should predict individual CWD status well because we expect that as herd CWD prevalence increases, individuals in the herd would be more likely to be exposed to the disease, and therefore test CWD+. Using herd prevalence as a null model allows us to assess the predictive power of adding variables of interest, such as PRNP genotype groups, to an already informative model. Our global model included four predictor variables: herd CWD prevalence, sex, codon 20 genotype and codon 225 genotype. CWD is generally more prevalent in males than females, likely because during breeding season, males roam widely, court many females and fight with other males, which all increase exposure risk [40,41]. Therefore, we included sex as a model covariate to distinguish the influence of PRNP genotypes from the influence of sex. Age can also influence the probability of testing CWD positive [40], but we did not have sufficient age data to include it as a covariate in our models. Lastly, to address potential spatial autocorrelation among herds in the three geographical regions of the state, we tested a model relating individual CWD status to geographical region. If region outperformed herd prevalence, this would suggest that individual disease status was based on some unmeasured variable related to these three geographical regions (e.g. neutral genetic structure), rather than due to disease prevalence in the herd. We performed model comparison among a total of eight models (table 2) using Akaike's information criterion (AIC) [42,43]. We used Pearson's correlation coefficients to ensure that model covariates were not collinear, considering coefficients greater than or equal to 0.7 to indicate collinearity [44]. For this analysis, we included all samples from focal herds that had both CWD status and genotype data available.

We then used Bayes factor to quantify the difference in the likelihood of individuals testing CWD− given the CWD− rate of each PRNP genotype. A Bayes factor equal to one would indicate no difference in the CWD− rate between genotypes, and in general, values 1–3 suggest poor evidence of a difference, values 3–10 suggest substantial evidence of a difference, values 10–100 suggest strong evidence of a difference and values greater than 100 suggest very strong evidence of a difference [45]. For this analysis, we used all genotyped samples.

## 2.5. Prediction 2: the slow allele will be more common in herds with longer exposure to CWD

With our second prediction, we evaluated evidence for disease-driven selection at the herd level by assessing the relationship between PRNP genotype frequencies and the relative length of time herds have been exposed to CWD. If selection was acting in this system, we would expect to observe higher

frequencies of the favoured slow allele in herds where selection has had more time to act (i.e. herds exposed to CWD for a longer time period). We calculated relative exposure time in each of our 14 focal herds by subtracting the year of first CWD detection from 2019, the final year of sample collection. We related codon 20 and 225 genotype frequencies in mule deer herds to the number of years since CWD was first detected using binomial regression with a complementary log–log (cloglog) link function in the R package *stats* v. 4.0.2 (table 2) [39]. A cloglog link function is appropriate for proportional response data with values clustered near zero [46].

## 2.6. Prediction 3: the slow allele will increase more over time in herds with higher CWD prevalence

With our third prediction, we further investigated evidence of disease-driven selection at the herd level by characterizing temporal changes in PRNP genotype frequencies in CWD-infected mule deer herds. If CWD was driving selection on the mule deer PRNP gene, we expected to observe an increase in slow alleles in infected herds and we expected to observe a larger increase in herds exposed to higher selective pressure. We first tested whether the slow 225*F genotype frequency significantly increased over a span of approximately 2 decades in seven of our focal herds. We compared current 225*F genotype frequencies to previously reported genotype frequencies from mule deer samples collected in 2001–2003 [15] using a paired *t*-test in the R package *stats* v. 4.0.2 [39].

After documenting a significant increase in 225*F genotype frequencies over time, we investigated whether herds exposed to higher CWD prevalence rates exhibited greater increases in 225*F genotype frequencies, with the idea that CWD prevalence might represent relative selection pressure imposed by the disease. We calculated the change in 225*F genotype frequencies by subtracting the previously reported genotype frequencies in each of these seven herds from our observed frequencies. We related the change in 225*F genotype frequencies to herd CWD prevalence using linear regression in the R package *stats* v. 4.0.2 [39]. We tested both the current CWD prevalence rates as well as CWD prevalence rates from 2001 to 2003 as predictor variables and compared models using AIC (table 2) [42,43]. PRNP codon 20 genotype frequencies were not reported for 2001–2003, so we excluded codon 20 from temporal analyses.

# 3. Results

## 3.1. PRNP sequencing and genotyping

We genotyped the PRNP gene in 1156 samples (661 males and 495 females). We found eight non-synonymous mutations and 15 synonymous mutations in the PRNP gene (electronic supplementary material, table S1), most of which were extremely rare (minor allele frequency less than 0.01). The minor allele at PRNP codon 20 (G) had a statewide frequency of 0.08 and the minor allele at PRNP codon 225 (F) had a statewide frequency of 0.04. Due to the rarity of minor alleles at codons 20 and 225, we used two genotype categories for each site in our analyses: codon 20DD (frequency 0.84), codon 20*G (frequency 0.16), codon 225SS (frequency 0.92) and codon 225*F (frequency 0.08).

## 3.2. Prediction 1: individuals possessing the slow allele will be less likely to test positive for CWD

Our global logistic regression model relating individual CWD status to herd CWD prevalence, individual sex, individual codon 20 genotype and individual codon 225 genotype performed best among the models we tested (table 2). Individual deer were more likely to test CWD+ if they were from a herd that had a higher CWD prevalence, if they were male, if they had a 20*G genotype, and if they had a 225SS genotype (figure 2). Herd prevalence had the most predictive power, followed by codon 255 genotype, sex and codon 20 genotype (figure 2c). For both sexes, 225*F individuals were unlikely to test CWD+ regardless of herd CWD prevalence, whereas 225SS individuals were more likely to test CWD+ as herd CWD prevalence increased (figure 2a; electronic supplementary material, figure S1). As herd CWD prevalence increased, the likelihood of testing CWD+ increased more for 225SS males than for 225SS females (electronic supplementary material, figure S1). For both sexes, 20DD individuals were slightly less likely to test CWD+, regardless of herd CWD prevalence. However, this relationship was not significant because 95% confidence intervals around the fitted curve overlapped considerably for the two codon 20 genotypes (figure 2b; electronic supplementary material, figure S1). Our second-best model excluded codon 20 genotype and performed similar to the global model (ΔAIC = 1.7), suggesting that codon 20

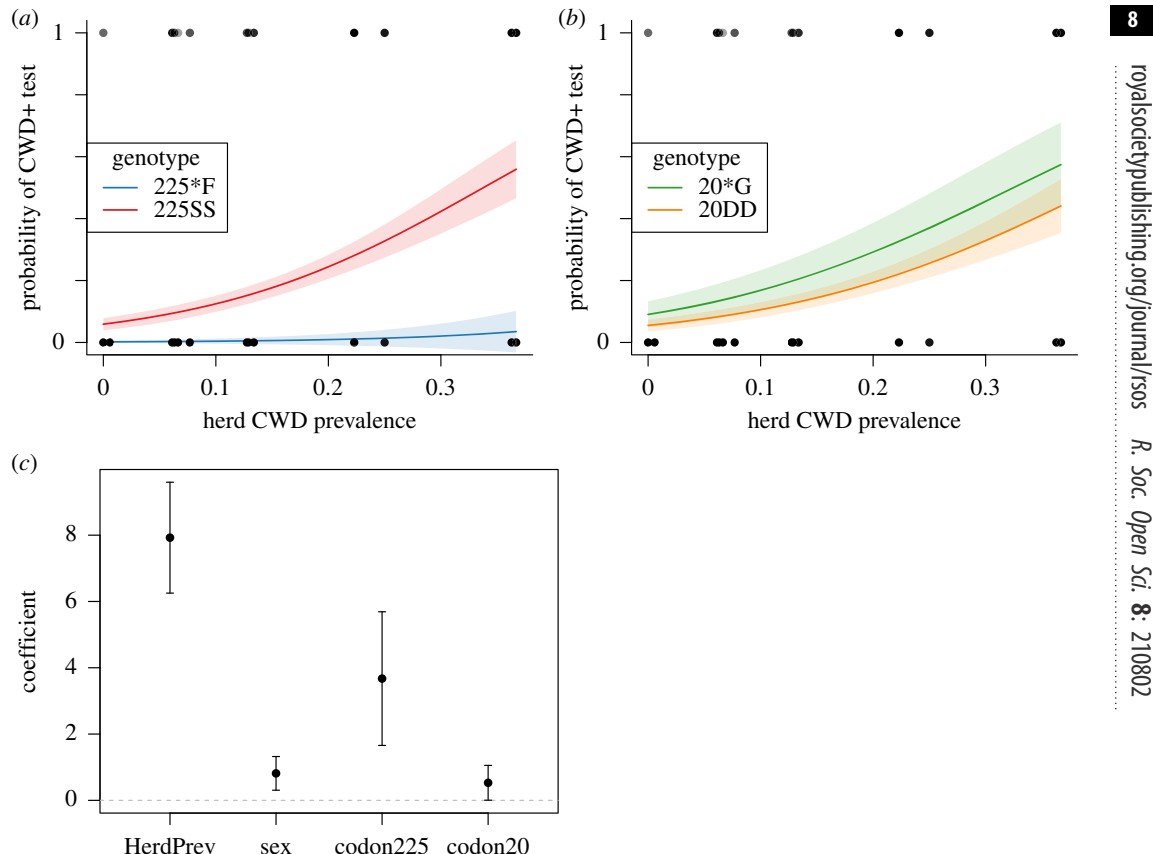

**Figure 2.** Logistic regression relating the probability of an individual deer testing CWD+ to herd CWD prevalence, sex, and either (*a*) PRNP codon 225 genotype or (*b*) PRNP codon 20 genotype. Black points represent individual deer test results (1 = CWD+, 0 = CWD−) plotted against CWD prevalence in the herd they were sampled in. Fitted model values (dark line) and 95% confidence interval around the fitted values (lighter shaded colour) are shown. $\beta$ coefficients with 95% confidence intervals are shown for the global model (*c*). See electronic supplementary material, figure S1 for each sex separately.

had little impact on the probability of testing CWD+. None of the covariates used in multivariate models had Pearson's correlation coefficient higher than 0.21. Lastly, our null model relating individual CWD status to herd prevalence outperformed our model relating individual CWD status to geographical region (table 2), indicating that the disease-specific variable of herd prevalence predicted individual disease status better than the spatial arrangement of herds in our study.

It was approximately 240 times more likely for 225*F individuals to test CWD−, given the 225*F CWD− rate than the 225SS CWD− rate, providing very strong evidence that the F allele reduced the likelihood of testing CWD+. It was approximately 2.7 times more likely for 20DD individuals to test CWD−, given the 20DD CWD− rate than the 225*G CWD− rate, providing poor evidence that the D allele affected the likelihood of testing CWD+.

### 3.3. Prediction 2: the slow allele will be more common in herds with longer exposure to CWD

The number of years since CWD was first detected in our 14 focal herds ranged from 2 to 27 years (table 1). We observed a positive relationship between the frequency of the 225*F genotype and the number of years since CWD was first detected in a herd ($\beta = 0.09 \pm 0.03$; figure 3*a*). The frequency of the 20DD genotype was unrelated to the number of years since CWD was first detected in a herd ($\beta = 0.007 \pm 0.006$; figure 3*b*).

### 3.4. Prediction 3: the slow allele will increase more over time in herds with higher CWD prevalence

The change in 225*F genotype frequencies between 2001–2003 and 2014–2019 in seven focal herds ranged from −0.01 to 0.24 (table 1). Overall, codon 225*F genotype frequencies significantly increased in the time

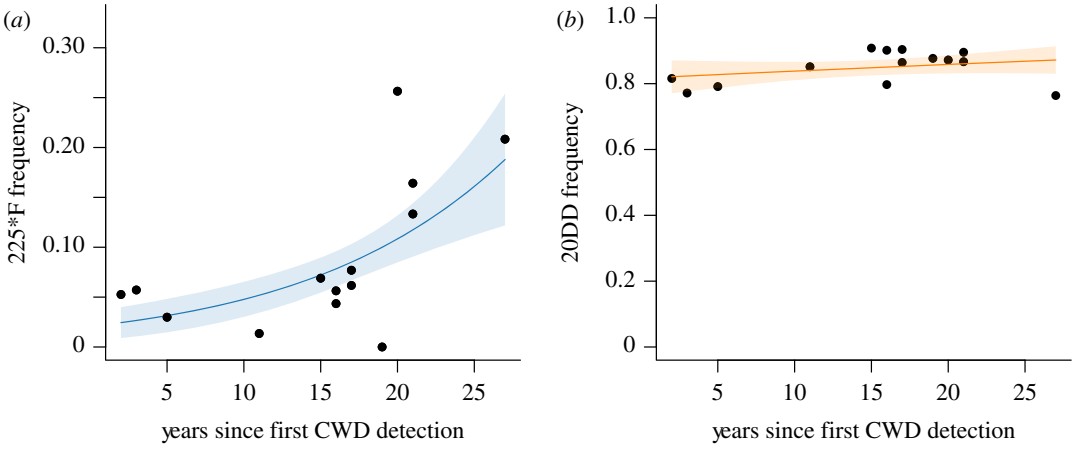

**Figure 3.** Binomial regression with a complementary log–log link function relating (*a*) PRNP codon 225*F genotype frequency and (*b*) PRNP codon 20DD genotype frequency to the number of years since CWD was first detected in a herd. Fitted values (dark line) and 95% confidence interval around the fitted values (lighter shaded colour) are shown. Based on 14 Wyoming mule deer herds sampled in 2014–2019.

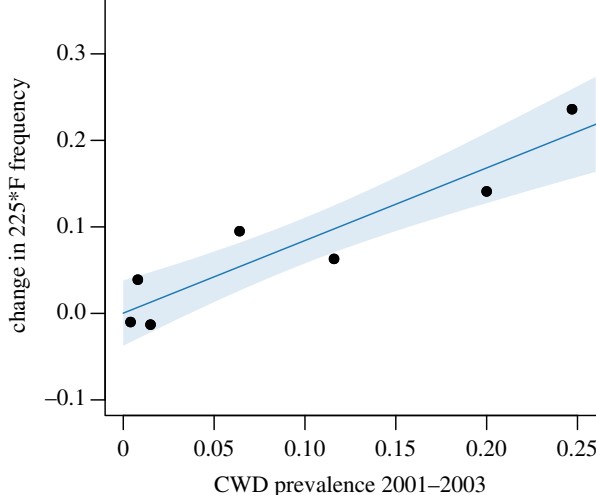

**Figure 4.** Linear regression relating the change in PRNP codon 225*F genotype frequency from 2001–2003 to 2014–2019 to herd CWD prevalence estimated in 2001–2003. Fitted values (dark line) and 95% confidence interval around the fitted values (lighter shaded colour) are shown. Based on seven mule deer herds in Wyoming sampled in both time periods.

between 2001–2003 and 2014–2019 (paired *t*-test, $p = 0.028$). Not only did 225*F genotype frequencies increase in less than 2 decades, 225*F increased more in herds with a higher CWD prevalence ($\beta = 0.84 \pm 0.15$; figure 4). The change in genotype frequencies was better represented by CWD prevalence rates estimated in 2001–2003 than CWD prevalence rates estimated in 2015–2019 ($\Delta\text{AIC} = 6.5$, table 2). Four mule deer herds aligned with our prediction because they increased to or maintained the highest CWD prevalence in the state and had the highest 225*F genotype frequencies in the state. By contrast, three herds started with the lowest prevalence rates in 2001–2003 and maintained relatively low prevalence and low 225*F genotype frequencies in 2015–2019 (table 1).

## 4. Discussion

We found support for all three of our predictions related to codon 225, including evidence that mule deer with the 225F allele were less likely to test positive for CWD, the 225F allele was more common in herds exposed to CWD longer and the 225F allele increased more over time in herds with higher CWD prevalence. If PRNP genotype is indeed directly related to disease phenotype in the form of a prolonged incubation period, infected deer with the slow 225F allele potentially have a longer lifespan

with the opportunity to produce more offspring of the same genotype, leading to selection favouring the slow 225F allele. This study adds to growing support for CWD-mediated selection on the PRNP gene in cervid species. Monello *et al.* [47] found that a slow genotype at codon 132 of the elk PRNP gene was more common in elk populations infected with CWD than uninfected populations. Robinson *et al.* [17] used age-specific CWD prevalence as a proxy for change over time to show lower infection rates and higher survival rates for white-tailed deer with a slow genotype at PRNP codon 96. Our study expanded on previous research by using both individual- and herd-based analyses and employing a novel evaluation of change in PRNP genotype frequencies over time to provide multiple lines of evidence for disease-driven selection in free-ranging mule deer in Wyoming, USA.

In contrast with our codon 225 results, we found little evidence that codon 20 relates to CWD status or herd prevalence. Although both mutations are non-synonymous, the final PrP protein product includes the amino acid encoded by codon 225 but not the amino acid encoded by codon 20, which is cleaved during membrane translocation [48]. Wilson *et al.* [31] suggested that if codon 20 itself does not affect disease phenotype, it may be linked to another mutation that is under selection. We found that genotypes at codons 20 and 225 were not strongly correlated, thus selection acting on codon 225 likely did not affect codon 20 genotype frequencies. It is possible that codon 20 was in linkage disequilibrium with another site outside the PRNP coding region that was subject to CWD-mediated selection [31], or that the slightly positive relationship with individual CWD status was spurious.

We observed the 225F allele throughout Wyoming, suggesting that selection favouring the F allele was acting on standing variation rather than a new mutation spreading concurrently with the expanding range of CWD. Adaptation from standing genetic variation can lead to faster evolution than a new mutation [49], which could explain the increase in 225*F genotype frequencies we observed in herds exposed to CWD for the past 2 decades, and could affect the pace of natural selection as CWD continues to reach new mule deer herds. Despite the apparent benefits of possessing the 225F allele, this mutation remains rare in mule deer herds. We observed only one individual with a 225FF genotype, and previous studies have similarly found few individuals with two copies of the slow F allele [15,19]. A clinical study that included 225FF mule deer documented atypical behaviour and poorer body condition in 225FF deer compared to 225SS deer [30]. If 225FF deer experience reduced fitness due to atypical characteristics, balancing selection could favour the heterozygous 225SF genotype in CWD-infected populations, as is often seen in MHC genes [50,51]. More data on the relative fitness of the three genotype groups (i.e. 225SS, 225SF, 225FF) is needed to distinguish between directional and balancing selection acting on codon 225 of the mule deer PRNP gene.

Our temporal comparison revealed three mule deer herds in southeastern Wyoming that exhibited relatively low CWD prevalence and low 225*F genotype frequencies despite being exposed to CWD for approximately 2 decades. It is unclear why these herds deviated from the predicted relationships we observed statewide, but it could be due to variation in relative selection pressure imposed by other ecological drivers (e.g. predation, hunting pressure, resource access, other diseases). For example, in the neighbouring state of Colorado, changes in mule deer CWD prevalence over time were attributed to variation in hunting pressure [52]. An investigation of genotype-specific fitness in these outlier mule deer herds compared to other herds in the state could illuminate the reason for this variation.

Population growth models for several cervid species have demonstrated the contribution of CWD to population declines, and the inclusion of genotype-specific variation in fitness metrics altered projected outcomes [17–19]. In mule deer, the population growth rate ($\lambda$) for a CWD-infected population in Wyoming was 0.79, but when the model accounted for genotype-specific survival rates, $\lambda$ decreased to 0.64 for 225SS deer and increased to 0.98 for 225*F deer [19]. DeVivo *et al.* demonstrated differential population growth rates based on PRNP genotypes but did not account for the expected change in PRNP genotype frequencies over time, which may alter the projected growth rate. Our study provides individual-level (probability of testing CWD+) and herd-level (expected PRNP codon 225 genotype frequency) models that could add natural selection dynamics to population growth models. For example, a population growth model that includes genotype-specific fitness metrics could also include current codon 225 genotype frequency (as measured in a population of interest) and expected change in genotype frequency (as predicted from our models). The model would account for CWD-driven selection by weighting genotype-specific fitness metrics according to the expected change in genotype frequency over time. Population growth models serve an important role in disease management by allowing managers to understand current disease dynamics and to predict future population growth rates under different management scenarios [18,53]. The usefulness of population models is a function of their ability to reasonably represent reality, so in a system where a disease drives natural selection, the inclusion of selection-related parameters should improve model predictions and consequently help to improve disease management.

Infectious diseases pose a threat to global biodiversity [1,2], and understanding how free-ranging wildlife populations respond to infectious diseases is essential for effective management and conservation [54]. Documentation of disease-mediated selection in free-ranging animal populations remains limited despite the significant selection pressure that infectious diseases can impose. Fitness variation associated with the cervid PRNP gene provides a valuable opportunity to investigate the role of infectious disease on selection in a natural system. In addition, as harvested species, cervids are the focus of extensive research and management efforts [53], which further extends the usefulness of CWD in cervids as a model of infectious diseases in wildlife populations. Future research can build on our study by examining evidence of CWD-driven selection in other mule deer populations, by expanding work on genotype-specific fitness in wild deer, and by building population growth models that incorporate metrics of selection derived from our research.

Data accessibility. PRNP sequences, sample metadata, and R code available from the Dryad Digital Repository: https://doi.org/10.5061/dryad.g79cnp5pq [55].

Authors' contributions. M.E.F.L. and H.B.E. conceptualized the study and developed methodology with help from W.H.E. H.B.E. provided laboratory resources and W.H.E. provided samples and disease data. H.B.E. acquired funding with help from M.E.F.L. H.B.E. supervised the project with help supervising laboratory work from M.E.F.L. and L.N.L.J. M.E.F.L. carried out the molecular laboratory work, formal analyses, data visualization and data validation with help from H.B.E. M.E.F.L. drafted the manuscript with help critically revising the manuscript from J.L.M., W.H.E., L.N.L.J., S.E.A. and H.B.E. All authors gave final approval for publication and agree to be held accountable for the work performed therein.

Competing interests. We declare we have no competing interests.

Funding. This work was supported by the Wyoming Governor's Big Game License Coalition. M.E.F.L. was supported by the Carlton R. Barkhurst Dissertation Fellowship, the University of Wyoming Program in Ecology, the Riverbend Endowment in Wildlife-Livestock Health and H.B.E's Wyoming Excellence Chair funds.

Acknowledgements. We thank the Wyoming Game and Fish Department for providing samples, data and expertise, including the WGFD Wildlife Health Lab (J.E. Jennings-Gaines, H. Killion, T. Stitzlein), C. Stewart and M.E. Wood. This work was facilitated by collaborations through the US Fish and Wildlife Service and the US Geological Survey under Grant/Cooperative Agreement No. G20AC00062 (P. Deibert, S. Oyler-McCance). We thank the Wyoming State Veterinary Laboratory, T.E. Cornish and M.T. DeVivo for their assistance. We thank S.M. Love Stowell for input on laboratory work, and thank E.G. Bentley, A.M. Mackenzie, M.D. Johnson and E. Winward for assistance in the laboratory. We thank C.A. Buerkle, M.A. Murphy, M.J. Kauffman and M. Ben-David for input on the manuscript.

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
