## [Peer Review File · Royal Society Open Science]

Review History

RSOS-210802.R0 (Original submission)

Review form: Reviewer 1

Is the manuscript scientifically sound in its present form?

Yes

Are the interpretations and conclusions justified by the results?

Yes

Is the language acceptable?

Yes

Do you have any ethical concerns with this paper?

No

Have you any concerns about statistical analyses in this paper?

Yes

Recommendation?

Accept with minor revision (please list in comments)

Comments to the Author(s)

This is a study of how variation in CWD susceptibility linked to the PRNP gene affect selection. Selection of PRNP genotypes relative to CWD status has been done previously a few times, but it is important to replicate such studies given their importance. The paper is very clear and well done. My comments are fairly minor apart from one statistical issue.

A main weakness of the study is that spatial structure was not taken into account in statistical analysis (as in many other studies). It is basically three regions driving these relationships. You should consider adding "area" as a random term in the models of individual data (e.g. LME4 package), and even better, some spatial correlation structure to account for spatial autocorrelation (more technically challenging).

Line 211. If you compare frequencies, why turn to t-test when you correctly use logistic regression elsewhere? (use e.g. the `cbind` command)

Minor comments

Line 312 onwards. What is the CWD prevalence in bordering Colorado hunting areas? I am thinking of the paper by Miller et al. 2020 in *J Wildl Dis* "Hunting pressure modulates...". A different management system for bucks in the areas? Just curious! Anything about the age composition in the herds? This can also affect CWD prevalence and also possible transmission rates.

Introduction first paragraph. Why focus on genomics, when you end up studying a single gene, PRNP?

Line 80. You should add that it was "in 2018" list of emerging issues.

Line 119 and table 1: Was CWD prevalence estimated separately for males and females? (as you state line 180, males are more often infected) How was age taken into account?

Line 124. As you correctly state, detection issues are important with CWD. Any information about sample sizes in the different herds? – could affect this timing variable.

Line 134. Could these genetics affect the PRNP composition in the first place? Should be possible to include in analysis of your 3rd prediction.

Line 265-268. This was not clear. "Our model based on herd prevalence..." -> be explicit on what model?

Line 280 "though they did not consider..." They did not explicitly consider, they did discuss this. Suggest to delete last part of sentence.

Line 326. Add reference after "This study" – it is not your study.

Conclusion is a bit generic.

Review form: Reviewer 2

Is the manuscript scientifically sound in its present form?

Yes

Are the interpretations and conclusions justified by the results?

Yes

Is the language acceptable?

Yes

Do you have any ethical concerns with this paper?

No

Have you any concerns about statistical analyses in this paper?

No

Recommendation?

Accept with minor revision (please list in comments)

Comments to the Author(s)

I enjoyed reading this manuscript on the impacts of CWD on mule deer genetics in Wyoming. The authors found that the 225F allele in mule deer is more common in areas with more CWD, increasing over time in those areas, and 225F individuals are less likely to be CWD positive. These results are in-line with expectations based on other studies, but the manuscript provides a wealth of new data to the important issue. I support the publication of this manuscript and have only a few minor suggestions for how the authors might further improve the manuscript.

Prediction 1:

1. Many readers will wonder why age is not included in the model. I may have missed it, but if age is available, then this would be an important covariate. If it is not available, then just state that it was not available.
2. Some may consider it a little odd to have the prevalence as a predictor of individual disease status. An slight alternative here, would be to remove herd prevalence, but add herd ID as a random effect. If some herds are more poorly sampled than others, than this structure would more appropriately account for that issue.
3. The authors could consider post-hoc an 225^*herd (or $225^*herdprevalence$) interaction term. The idea here being that at low prevalence there may be limited impact of the 225 allele, but that impact increases with herd prevalence.
4. The authors should state or discuss their proposed mechanism for prediction 1. I may have missed it, but one mechanism is the longer lag time between exposure and becoming test positive. An alternative (but less likely) is that the exposure rate differs. A third alternative is that 225F's may survive longer with CWD and thus be at higher prevalence than others. The last one is not supported by the data, but going through all the alternatives seems helpful in the intro or discussion.

Prediction 2 & 3. It is a little unclear here what models were run and their exact structure. I suggest writing the best model out explicitly or including a table of all the options that were run. In predictions 2 & 3 we have genotype as the dependent variable and CWD as a predictor, whereas Prediction 1 was the reverse direction. In terms of mechanisms, the focus here is on how CWD may alter genotype frequency, so having genotype as the dependent variable makes sense. Given the mechanism of a lag before being test positive I can see why the authors did the reverse

for #1, but I think an explanation would help the reader, and maybe a re-ordering such that Prediction 1 gets dropped to the 3rd position.

I thought the discussion around the potential costs of being 225FF and integrating natural selection into population models was really good.

Decision letter (RSOS-210802.R0)

Dear Dr Ernest

On behalf of the Editors, we are pleased to inform you that your Manuscript RSOS-210802 "Spatiotemporal analyses reveal infectious disease-driven selection in a free-ranging ungulate" has been accepted for publication in Royal Society Open Science subject to minor revision in accordance with the referees' reports. Please find the referees' comments along with any feedback from the Editors below my signature.

Please submit your revised manuscript and required files (see below) no later than 7 days from today's (ie 14-Jun-2021) date. Note: the ScholarOne system will 'lock' if submission of the revision is attempted 7 or more days after the deadline. If you do not think you will be able to meet this deadline please contact the editorial office immediately.

on behalf of Dr Joachim Mergeay (Associate Editor) and Kevin Padian (Subject Editor)
openscience@royalsociety.org

Associate Editor Comments to Author (Dr Joachim Mergeay):

Dear authors,

We have received two promising reviews of your manuscript. Both reviewers provide constructive criticism and suggestions with regards to the statistical analyses (most important is about independence, identified by rev1), most of which are relatively easy to tackle and integrate in the current methodology.

We would be happy to see a revised version that takes these comments into account.

Best regards,
J Mergeay

Reviewer comments to Author:

Reviewer: 1

Comments to the Author(s)

This is a study of how variation in CWD susceptibility linked to the PRNP gene affect selection. Selection of PRNP genotypes relative to CWD status has been done previously a few times, but it is important to replicate such studies given their importance. The paper is very clear and well done. My comments are fairly minor apart from one statistical issue.

A main weakness of the study is that spatial structure was not taken into account in statistical analysis (as in many other studies). It is basically three regions driving these relationships. You should consider adding "area" as a random term in the models of individual data (e.g. LME4 package), and even better, some spatial correlation structure to account for spatial autocorrelation (more technically challenging).

Line 211. If you compare frequencies, why turn to t-test when you correctly use logistic regression elsewhere? (use e.g. the `cbind` command)

Minor comments

Line 312 onwards. What is the CWD prevalence in bordering Colorado hunting areas? I am thinking of the paper by Miller et al. 2020 in *J Wildl Dis* "Hunting pressure modulates..". A different management system for bucks in the areas? Just curious! Anything about the age composition in the herds? This can also affect CWD prevalence and also possible transmission rates.

Introduction first paragraph. Why focus on genomics, when you end up studying a single gene, PRNP?

Line 80. You should add that it was "in 2018" list of emerging issues.

Line 119 and table 1: Was CWD prevalence estimated separately for males and females? (as you state line 180, males are more often infected) How was age taken into account?

Line 124. As you correctly state, detection issues are important with CWD. Any information about sample sizes in the different herds? - could affect this timing variable.

Line 134. Could these genetics affect the PRNP composition in the first place? Should be possible to include in analysis of your 3rd prediction.

Line 265-268. This was not clear. "Our model based on herd prevalence..." -> be explicit on what model?

Line 280 “though they did not consider...” They did not explicitly consider, they did discuss this. Suggest to delete last part of sentence.

Line 326. Add reference after “This study” – it is not your study.

Conclusion is a bit generic.

Reviewer: 2

Comments to the Author(s)

I enjoyed reading this manuscript on the impacts of CWD on mule deer genetics in Wyoming.

The authors found that the 225F allele in mule deer is more common in areas with more CWD, increasing over time in those areas, and 225F individuals are less likely to be CWD positive.

These results are in-line with expectations based on other studies, but the manuscript provides a wealth of new data to the important issue. I support the publication of this manuscript and have only a few minor suggestions for how the authors might further improve the manuscript.

Prediction 1:

1. Many readers will wonder why age is not included in the model. I may have missed it, but if age is available, then this would be an important covariate. If it is not available, then just state that it was not available.

2. Some may consider it a little odd to have the prevalence as a predictor of individual disease status. A slight alternative here, would be to remove herd prevalence, but add herd ID as a random effect. If some herds are more poorly sampled than others, than this structure would more appropriately account for that issue.

3. The authors could consider post-hoc an 225*herd (or 225*herdprevalence) interaction term. The idea here being that at low prevalence there may be limited impact of the 225 allele, but that impact increases with herd prevalence.

4. The authors should state or discuss their proposed mechanism for prediction 1. I may have missed it, but one mechanism is the longer lag time between exposure and becoming test positive. An alternative (but less likely) is that the exposure rate differs. A third alternative is that 225F's may survive longer with CWD and thus be at higher prevalence than others. The last one is not supported by the data, but going through all the alternatives seems helpful in the intro or discussion.

Prediction 2 & 3. It is a little unclear here what models were run and their exact structure. I suggest writing the best model out explicitly or including a table of all the options that were run. In predictions 2 & 3 we have genotype as the dependent variable and CWD as a predictor, whereas Prediction 1 was the reverse direction. In terms of mechanisms, the focus here is on how CWD may alter genotype frequency, so having genotype as the dependent variable makes sense. Given the mechanism of a lag before being test positive I can see why the authors did the reverse for #1, but I think an explanation would help the reader, and maybe a re-ordering such that Prediction 1 gets dropped to the 3rd position.

I thought the discussion around the potential costs of being 225FF and integrating natural selection into population models was really good.

===PREPARING YOUR MANUSCRIPT===

===PREPARING YOUR REVISION IN SCHOLARONE===

- Any electronic supplementary material (ESM).
- If you are requesting a discretionary waiver for the article processing charge, the waiver form must be included at this step.
- If you are providing image files for potential cover images, please upload these at this step, and inform the editorial office you have done so. You must hold the copyright to any image provided.
- A copy of your point-by-point response to referees and Editors. This will expedite the preparation of your proof.

- Ensure that your data access statement meets the requirements at <https://royalsociety.org/journals/authors/author-guidelines/#data>. You should ensure that you cite the dataset in your reference list. If you have deposited data etc in the Dryad repository, please only include the 'For publication' link at this stage. You should remove the 'For review' link.
- If you are requesting an article processing charge waiver, you must select the relevant waiver option (if requesting a discretionary waiver, the form should have been uploaded at Step 3 'File upload' above).
- If you have uploaded ESM files, please ensure you follow the guidance at <https://royalsociety.org/journals/authors/author-guidelines/#supplementary-material> to include a suitable title and informative caption. An example of appropriate titling and captioning may be found at https://figshare.com/articles/Table_S2_from_Is_there_a_trade-off_between_peak_performance_and_performance_breadth_across_temperatures_for_aerobic_scope_in_teleost_fishes_/3843624.

Author's Response to Decision Letter for (RSOS-210802.R0)

See Appendix A.

Decision letter (RSOS-210802.R1)

Dear Dr Ernest,

I am pleased to inform you that your manuscript entitled "Spatiotemporal analyses reveal infectious disease-driven selection in a free-ranging ungulate" is now accepted for publication in Royal Society Open Science.

If you have not already done so, please remember to make any data sets or code libraries 'live' prior to publication, and update any links as needed when you receive a proof to check - for

instance, from a private 'for review' URL to a publicly accessible 'for publication' URL. It is good practice to also add data sets, code and other digital materials to your reference list.

You can expect to receive a proof of your article in the near future. Please contact the editorial office (openscience@royalsociety.org) and the production office (openscience_proofs@royalsociety.org) to let us know if you are likely to be away from e-mail contact – if you are going to be away, please nominate a co-author (if available) to manage the proofing process, and ensure they are copied into your email to the journal. Due to rapid publication and an extremely tight schedule, if comments are not received, your paper may experience a delay in publication.

on behalf of Dr Joachim Mergeay (Associate Editor) and Kevin Padian (Subject Editor)
openscience@royalsociety.org

Appendix A

UNIVERSITY
OF WYOMING

Department of Veterinary Sciences

WILDLIFE GENOMICS
& DISEASE ECOLOGY

Corresponding author:

Holly B. Ernest

1174 Snowy Range Rd

Laramie, WY, 82070

holly.ernest@uwyo.edu

June 23, 2021

To Dr. Jeremy Sanders, Editor-in-Chief, *Royal Society Open Science*:

Please find a revision of our manuscript entitled, 'Spatiotemporal analyses reveal infectious disease-driven selection in a free-ranging ungulate,' by Melanie E. F. LaCava, Jennifer L. Malmberg, William H. Edwards, Laura N.L. Johnson, Samantha E. Allen, and Holly B. Ernest, for publication as a research article in *Royal Society Open Science*.

In the original manuscript, now revised, we investigated natural selection in wild mule deer in relation to chronic wasting disease (CWD), which has gradually spread across Wyoming, USA, creating natural variation in disease history for our study. We characterized the relationship between CWD and a mutation in the mule deer prion protein gene that slows disease progression using 1,156 deer sampled across multiple ecosystems. We used both individual- and herd-based analyses and employed a novel temporal comparison to provide multiple lines of evidence for disease-driven selection in free-ranging mule deer. Our study provides new information to incorporate natural selection into population models to better predict future disease dynamics.

The manuscript received favorable and constructive reviews that guided our revision and resulted in an improved manuscript. Important revisions to the manuscript include the following:

1. We introduced an additional model to our first prediction to address potential spatial autocorrelation
2. We improved our explanation of the logic driving each of our three predictions
3. We clarified descriptions of our data and definitions of our models

We made additional minor revisions based on recommendations from reviewers. Below you will find documentation of all changes we made to the manuscript and we have noted point-by-point, with reference to content and line numbers, how we addressed each of the reviewers' comments specifically. We have included in our submission electronic supplementary materials and a link to a Dryad Digital Repository containing data and code to replicate our analyses in full. The enclosed work is not under consideration for publication in another journal. All authors have read and agreed to the submission of this version of the manuscript. The authors have no conflicts of interest to declare.

Thank you for the opportunity to revise this manuscript for publication,

Holly Ernest, DVM, PhD (corresponding author) & Melanie LaCava, PhD (first author)

Line-by-line responses to reviewer comments:

(Please note that our responses to individual reviewer comments each begin with RESPONSE and are bolded to help distinguish them from the reviews. Line number listed here correspond to the “track changes” version of the manuscript.)

Associate Editor Comments to Author (Dr Joachim Mergeay):

Dear authors,

We have received two promising reviews of your manuscript. Both reviewers provide constructive criticism and suggestions with regards to the statistical analyses (most important is about independence, identified by rev1), most of which are relatively easy to tackle and integrate in the current methodology.

We would be happy to see a revised version that takes these comments into account.

Best regards,

J Mergeay

Reviewer comments to Author:

Reviewer: 1

Comments to the Author(s)

This is a study of how variation in CWD susceptibility linked to the PRNP gene affect selection. Selection of PRNP genotypes relative to CWD status has been done previously a few times, but it is important to replicate such studies given their importance. The paper is very clear and well done. My comments are fairly minor apart from one statistical issue.

A main weakness of the study is that spatial structure was not taken into account in statistical analysis (as in many other studies). It is basically three regions driving these relationships. You should consider adding “area” as a random term in the models of individual data (e.g. LME4 package), and even better, some spatial correlation structure to account for spatial autocorrelation (more technically challenging).

RESPONSE: To address the concern about spatial autocorrelation among herds in our study, we ran a model relating individual CWD status to geographic region (CWD+ ~ Region) to compare with our current null model (CWD+ ~ HerdPrev). If region outperformed herd prevalence, this would suggest that individual disease status was based on some unmeasured variable related to these three geographic regions (e.g., neutral genetic structure), rather than due to variation in disease prevalence within the herd. Geographic region performed worse than herd prevalence as a predictor (AIC= 644.5 for CWD+ ~ Region vs. AIC=640.7 for CWD+ ~ HerdPrev), suggesting that herd prevalence represents the observed

pattern better than spatial structure. We did not try including geographic region as a random effect in our models because Gelman and Hill 2006 (doi:10.1017/CBO9780511790942) recommend against using random effects with less than five groups, and also the geographic regions are not truly random. We incorporated the additional tested model into our paper in three places: in table 2; in lines 199–203 of the methods section, “Lastly, to address potential spatial autocorrelation among herds in the three geographic regions of the state, we tested a model relating individual CWD status to geographic region. If region outperformed herd prevalence, this would suggest that individual disease status was based on some unmeasured variable related to these three geographic regions (e.g., neutral genetic structure), rather than due to disease prevalence in the herd;” and in lines 280–283 of the results section, “Lastly, our null model relating individual CWD status to herd prevalence outperformed our model relating individual CWD status to geographic region ($\Delta AIC = 48.7$), indicating that the disease-specific variable of herd prevalence predicted individual disease status better than the spatial clumping of herds in our study.”

Line 211. If you compare frequencies, why turn to t-test when you correctly use logistic regression elsewhere? (use e.g. the `cbind` command)

RESPONSE: The t-test and linear regression serve different purposes for our third prediction. We initially performed a t-test to determine whether the frequency of individuals possessing the 225F allele increased over time. If we had not observed a significant increase in genotype frequency, we likely would not have proceeded with our linear regression to assess the relationship between an increase in genotype frequency and herd CWD prevalence. We do not believe a logistic regression would serve the same purpose as the t-test in this case. We made the purpose of the t-test clearer in our methods section with this updated excerpt on lines 236–244, “We first tested whether the slow 225*F genotype frequency significantly increased over a span of approximately two decades in seven of our focal herds. We compared current 225*F genotype frequencies to previously reported genotype frequencies from mule deer samples collected in 2001–2003 using a paired t-test in the R package `stats` version 4.0.2. After documenting a significant increase in genotype frequencies over time, we investigated whether herds exposed to higher CWD prevalence rates exhibited greater increases in 225*F genotype frequencies, with the idea that CWD prevalence might represent relative selection pressure imposed by the disease.”

Minor comments

Line 312 onwards. What is the CWD prevalence in bordering Colorado hunting areas? I am thinking of the paper by Miller et al. 2020 in *J Wildl Dis* “Hunting pressure modulates...”. A different management system for bucks in the areas? Just curious! Anything about the age composition in the herds? This can also affect CWD prevalence and also possible transmission rates.

RESPONSE: Thank you for the suggested reference. We added the following note about the Miller et al. 2020 paper to lines 358–359 of this discussion paragraph as an example of what could be driving the unexplained variation we observed, “For example, in the neighboring state of Colorado, changes in mule deer CWD prevalence over time were attributed to variation in hunting pressure [52].”

Introduction first paragraph. Why focus on genomics, when you end up studying a single gene, PRNP?

RESPONSE: We reworded the first paragraph to reduce emphasis on genomics, so it now reads, “Infectious diseases pose a significant threat to global biodiversity and require extensive research and resources to manage [1–3]. Documenting selection pressure imposed by infectious diseases in natural systems remains challenging due to complex disease-host relationships and the myriad factors influencing host fitness. Previous research has demonstrated a relationship between wildlife disease phenotypes and diversity in immune-related genes such as the major histocompatibility complex (MHC) [4–6], with less focus on variation in other genes [7]. Methods such as genome-wide association studies are commonly applied to identify putative loci under selection [8], but the identification of mechanistic links between disease phenotypes and specific mutations in the genome remains limited in free-ranging animal populations [7]. Chronic wasting disease (CWD) is an infectious prion disease in captive and free-ranging cervids for which mutations in a single gene have been linked to variation in disease outcomes in affected species [9], providing a rare opportunity to study disease-driven selection in free-ranging wildlife.”

Line 80. You should add that it was “in 2018” list of emerging issues.

RESPONSE: We added the year as suggested.

Line 119 and table 1: Was CWD prevalence estimated separately for males and females? (as you state line 180, males are more often infected) How was age taken into account?

RESPONSE: We estimated CWD prevalence for each herd based on a combination of males and females because we wanted to compare current prevalence with historical prevalence, and the source for historical CWD data did not provide separate estimates for each sex. To make this clearer in our methods, we added the following sentence to lines 120–122, “We used a single CWD prevalence estimate based on both sexes combined for each herd so that our current prevalence estimates would be comparable with historical estimates that did not specify the sex of sampled deer.” We did not have sufficient age data available to include it as a covariate in our modeling. We added the following sentence to our methods on lines 197–199 to make this clear, “Age can also influence the probability of testing CWD positive [40], but we did not have sufficient age data to include it as a covariate in our models.”

Line 124. As you correctly state, detection issues are important with CWD. Any information about sample sizes in the different herds? – could affect this timing variable.

RESPONSE: We report sample sizes included in CWD prevalence estimates from 2015–2019 in Table 1. Regarding the timing variable specifically, CWD surveillance efforts did not begin in the same year for every herd in the state, so our metric of number of years since first detection of CWD represents a rough comparison of length of exposure to CWD in different herds. We made this clearer in our methods with the following sentence on lines 126–132, “CWD surveillance began in different years in different herds, so year of first detection is not a perfect corollary for the year CWD first arrived in each herd; however, surveillance expanded throughout the state as CWD spread, so the year of first detection represents an approximation for the relative length of time that herds have been exposed to the disease.”

Line 134. Could these genetics affect the PRNP composition in the first place? Should be possible to include in analysis of your 3rd prediction.

RESPONSE: As mentioned above, we addressed the reviewer’s concern about potential spatial autocorrelation by testing geographic region as a predictor variable for individual CWD status, and we found that the model with geographic region performed worst among the models tested (see table 2). By testing whether geographic region predicted individual disease status better than other covariates, we were essentially testing whether any variation among the three geographic regions affected our findings, and this includes neutral genetic structure. We noted genetic structure as an example of a spatial variable that could have led geographic region to predict individual disease status well in our methods on lines 201–203, “If region outperformed herd prevalence, this would suggest that individual disease status was based on some unmeasured variable related to these three geographic regions (e.g., neutral genetic structure), rather than due to disease prevalence in the herd.”

Line 265-268. This was not clear. “Our model based on herd prevalence...” - > be explicit on what model?

RESPONSE: We simplified this sentence to make it more clear and pointed readers to our updated table 2 that now explicitly defines the models tested: “The change in genotype frequencies was better represented by CWD prevalence rates estimated in 2001–2003 than CWD prevalence rates estimated in 2015–2019 ($\Delta AIC = 6.5$, table 2).”

Line 280 “though they did not consider...” They did not explicitly consider, they did discuss this. Suggest to delete last part of sentence.

RESPONSE: We deleted the last part of the sentence as suggested.

Line 326. Add reference after “This study” – it is not your study.

RESPONSE: We replaced “This study” with “DeVivo et al.” to reference the authors of the cited work more clearly.

Conclusion is a bit generic.

RESPONSE: Reviewer #2 and we believe the discussion sufficiently describes the outcome and relevance of our study and how it fits in with existing research. However, we added a sentence to lines 314–317 to reiterate the proposed mechanism of selection for our study, “If PRNP genotype is indeed directly related to disease phenotype in the form of a prolonged incubation period, infected deer with the slow 225F allele potentially have a longer life span with the opportunity to produce more offspring of the same genotype, leading to selection favoring the slow 225F allele.” In addition, we added a more relevant and more recent example of variation in selection pressure to lines 358–359, “For example, in the neighboring state of Colorado, changes in mule deer CWD prevalence over time were attributed to variation in hunting pressure [52].”

Reviewer: 2

Comments to the Author(s)

I enjoyed reading this manuscript on the impacts of CWD on mule deer genetics in Wyoming. The authors found that the 225F allele in mule deer is more common in areas with more CWD, increasing over time in those areas, and 225F individuals are less likely to be CWD positive. These results are in-line with expectations based on other studies, but the manuscript provides a wealth of new data to the important issue. I support the publication of this manuscript and have only a few minor suggestions for how the authors might further improve the manuscript.

Prediction 1:

1. Many readers will wonder why age is not included in the model. I may have missed it, but if age is available, then this would be an important covariate. If it is not available, then just state that it was not available.

RESPONSE: We did not have sufficient age data available to include it as a covariate in our modeling. We added the following sentence to our methods on lines 197–199 to make this clear, “Age can also influence the probability of testing CWD positive [40], but we did not have sufficient age data to include it as a covariate in our models.”

2. Some may consider it a little odd to have the prevalence as a predictor of individual disease status. A slight alternative here, would be to remove herd prevalence, but add

herd ID as a random effect. If some herds are more poorly sampled than others, than this structure would more appropriately account for that issue.

RESPONSE: We used CWD prevalence as the predictor in our null model so that we started off with an informative null (as CWD prevalence increases in a herd, we expect more individuals to test CWD+) and tested whether adding other covariates improved the explanatory power of the model. We made this clearer in our methods by adding the following text to lines 188–192, “Herd CWD prevalence should predict individual CWD status well because we expect that as herd CWD prevalence increases, individuals in the herd would be more likely to be exposed to the disease, and therefore test CWD+. Using herd prevalence as a null model allows us to assess the predictive power of adding variables of interest, such as PRNP genotype groups, to an already informative model.”

3. The authors could consider post-hoc an 225*herd (or 225*herdprevalence) interaction term. The idea here being that at low prevalence there may be limited impact of the 225 allele, but that impact increases with herd prevalence.

RESPONSE: This is an interesting idea. We tested adding an interaction term to all of our models that included both herd prevalence and codon 225 genotype, however these models performed worse than the same models without the interaction term (see AIC comparison below), so we left the models in the manuscript unchanged.

Model	AIC	AIC without interaction term
CWD+ ~ HerdPrev * codon225 + sex + codon20	596.9	595.8
CWD+ ~ HerdPrev * codon225 + sex	598.6	597.5
CWD+ ~ HerdPrev * codon225	607.6	606.4

4. The authors should state or discuss their proposed mechanism for prediction 1. I may have missed it, but one mechanism is the longer lag time between exposure and becoming test positive. An alternative (but less likely) is that the exposure rate differs. A third alternative is that 225F's may survive longer with CWD and thus be at higher prevalence than others. The last one is not supported by the data, but going through all the alternatives seems helpful in the intro or discussion.

RESPONSE: We added our proposed mechanism for prediction 1 on lines 183–186, as follows, “We predicted that individuals possessing the slow allele will be less likely to test positive for CWD because one aspect of slower disease progression may be a longer lag time between exposure to CWD and accumulation of detectable levels of infectious prions in deer with the slow allele.”

Prediction 2 & 3. It is a little unclear here what models were run and their exact

structure. I suggest writing the best model out explicitly or including a table of all the options that were run.

RESPONSE: We expanded table 2 so that it clearly lists the models we tested for all three of our predictions. The expansion includes an additional column to indicate which of our three predictions the listed models correspond to, and additional rows for prediction 2 and 3 models. The updated table 2 and caption is copied below.

Table 2. List of models tested for each of our three predictions. For our first prediction we used logistic regression to relate individual CWD status (CWD+ or CWD-) to herd CWD prevalence (HerdPrev), sex, PRNP codon 225 genotype (codon225), and PRNP codon 20 genotype (codon20). For our second prediction, we used binomial regression with a complementary log-log link function to relate the frequency of the 225*F genotype (225*F) to the year CWD was first detected in each herd (YrsSinceDetect). For our third prediction, we used linear regression to relate the change in frequency of the 225*F genotype from 2001–2003 to 2015–2019 (Change225*F) to herd CWD prevalence calculated in 2001–2003 (HerdPrev_01_03) or in 2015–2019 (HerdPrev_15_19). We report AIC and ΔAIC if we performed model comparison.

Prediction	Models	AIC	ΔAIC
#1	CWD+ ~ HerdPrev + sex + codon225 + codon20	595.8	0
	CWD+ ~ HerdPrev + sex + codon225	597.5	1.7
	CWD+ ~ HerdPrev + codon225	606.4	10.6
	CWD+ ~ HerdPrev + sex + codon20	632.3	36.5
	CWD+ ~ HerdPrev + sex	634.3	38.5
	CWD+ ~ HerdPrev + codon20	638.6	42.8
	CWD+ ~ HerdPrev	640.7	44.9
	CWD+ ~ Region	644.5	48.7
#2	225*F ~ YrsSinceDetect	-	-
	20DD ~ YrsSinceDetect	-	-
#3	Change225*F ~ HerdPrev_01_03	-23.2	0
	Change225*F ~ HerdPrev_15_19	-16.8	6.6

In predictions 2 & 3 we have genotype as the dependent variable and CWD as a predictor, whereas Prediction 1 was the reverse direction. In terms of mechanisms, the focus here is on how CWD may alter genotype frequency, so having genotype as the dependent variable makes sense. Given the mechanism of a lag before being test positive I can see why the authors did the reverse for #1, but I think an explanation would help the reader, and maybe a re-ordering such that Prediction 1 gets dropped to the 3rd position.

RESPONSE: We added more explanation to the methods section for all three of our predictions to help the readers keep track of the logic behind each one. For prediction #1 on lines 180–186, we now explain, “With our first prediction we

evaluated whether an individual's disease status was related to their PRNP genotype at codon 20 or codon 225. We predicted that individuals possessing the slow allele at either site will be less likely to test positive for CWD because one aspect of slower disease progression may be a longer lag time between exposure to CWD and accumulation of detectable levels of infectious prions in deer with the slow allele [9].” For prediction #2 on lines 216–220, we now explain, “With our second prediction we evaluated evidence for disease-driven selection at the herd level by assessing the relationship between PRNP genotype frequencies and the relative length of time herds have been exposed to CWD. If selection was acting in this system, we would expect to observe higher frequencies of the favored slow allele in herds where selection has had more time to act (i.e., herds exposed to CWD for a longer time period).” And for prediction #3 on lines 230–234, we now explain, “With our third prediction we further investigated evidence of disease-driven selection at the herd level by characterizing temporal changes in PRNP genotype frequencies in CWD-infected mule deer herds. If CWD was driving selection on the mule deer PRNP gene, we expected to observe an increase in slow alleles in infected herds and we expected to observe a larger increase in herds exposed to higher selective pressure.”

I thought the discussion around the potential costs of being 225FF and integrating natural selection into population models was really good.